# Intracellular Redox Behavior of Quercetin and Resveratrol Singly and in Mixtures

**DOI:** 10.3390/molecules28124682

**Published:** 2023-06-09

**Authors:** Maria Elena Giordano, Maria Giulia Lionetto

**Affiliations:** 1Department of Biological and Environmental Sciences and Technologies (DiSTeBA), University of Salento, 73100 Lecce, Italy; elena.giordano@unisalento.it; 2NBFC, National Biodiversity Future Center, 90133 Palermo, Italy

**Keywords:** quercetin, resveratrol, mixture, antioxidant, pro-oxidant, CM-H_2_DCFDA

## Abstract

Polyphenols have attracted great interest as potent antioxidant compounds and nutraceuticals; however, their antioxidant properties represent a multifaceted phenomenon, including pro-oxidant effects under particular conditions and complex behavior when multiple polyphenols are simultaneously present. Moreover, their intracellular behavior cannot always be predicted from their ability to counteract the production of ROS in acellular assays. The present work aimed to study the direct intracellular redox activity of two polyphenols, resveratrol and quercetin, singly and in mixture in a cellular short-term bioassay under both basal and pro-oxidant conditions. The study was carried out by spectrofluorimetric assessment of the intracellular fluorescence of CM-H_2_DCFDA-charged HeLa cells under either basal conditions, due to the reactive species associated with the normal cellular oxidative metabolism, or pro-oxidant conditions induced by H_2_O_2_ exposure. Under basal conditions, the obtained results showed a significant antioxidant effect of quercetin and a weaker antioxidant effect of resveratrol when used singly, while antagonism of their effect was detected in their equimolar mixtures at all the concentrations used. Under exposure of the cells to H_2_O_2_, quercetin exhibited a dose-dependent intracellular antioxidant activity whereas resveratrol manifested a pro-oxidant intracellular activity, while their equimolar mixtures showed an intracellular interaction between the 2 polyphenols, with additive effects at 5 µM and synergic at 25 µM and 50 µM. Thus, the results clarified the direct intracellular antioxidant/pro-oxidant activity of quercetin and resveratrol alone and in their equimolar mixtures in the cell model HeLa cells and highlighted that the antioxidant properties of polyphenols in mixtures at the cellular level depend not only on the nature of the compounds themselves but also on the type of interactions in the cellular system, which in turn are influenced by the concentration and the oxidative status of the cell.

## 1. Introduction

Oxidative stress is recognized as an important factor in the genesis of several pathological conditions, particularly chronic and degenerative diseases [1]. This justifies the great interest in antioxidant compounds and their protective effects against oxidative damage. In particular, polyphenols have attracted great interest in the last decades as potent antioxidants; they have catalyzed attention in the research field of functional foods and nutraceuticals within the wide research area of food chemistry. Naturally occurring polyphenols are the most abundant group of antioxidant compounds in the human diet. This class of molecules, which has more than 8000 structural variants, includes various secondary metabolites of plants characterized by the presence of aromatic rings with 1 or more phenolic functionalities [2]. Their antioxidant activity can be expressed through several pathways involving the direct capacity to scavenge reactive oxygen species (ROS) [3,4] and, in addition, the ability to stimulate the biosynthesis of antioxidant enzymes [5] and to modulate the expression and/or activity of ROS-producing enzymes [6] and endogenous antioxidant-synthesizing enzymes [7]. Their direct antioxidant activity is related to reactions with ROS and reactive nitrogen species to produce unreactive products and the chelation of transition metal ions to prevent in turn hydroxyl radical (^•^OH) formation [8].

Polyphenols are classified into flavonoids and nonflavonoids. Flavonoids share a common structure formed by two aromatic rings, indicated as A and B, linked together by three carbon atoms forming an oxygenated heterocycle, the C ring. They show a high antiradical activity and transition metal chelation activity, particularly those, such as quercetin, characterized by the presence of a catechol group (3′,4′-dihydroxy) in the B ring, a double bond between carbon 2 and carbon 3 of the C ring conjugated with the keto group at position 4, and a hydroxyl group in position 3 of ring C or in the carbon 5 of the A ring [9] (Figure 1). Nonflavonoids represent a heterogeneous group including phenolic acids, stilbenes, and lignans as the main subgroups. Resveratrol is one of the most known nonflavonoid polyphenols, characterized like all the other stilbenes by 2 phenyl moieties linked together by a 2-carbon methylene group with phenyl groups substituted at positions 3, 5, and 4′ by hydroxy groups (Figure 1).

The antioxidant properties of polyphenols represent a multifaceted phenomenon, and some aspects deserve particular attention. For instance, although the antioxidant properties of polyphenols are well recognized, some pro-oxidant effects have also been described as often associated with their proapoptotic activity in various cell types [10]. Phenolic compounds are known to act as pro-oxidants under certain conditions, depending on concentration, pH, and the presence of transition metals [11,12]. Their pro-oxidant activity is based on the formation of a labile aroxyl radical that can react with oxygen, forming superoxide ions [13]. In the presence of transition metals, their pro-oxidant activity is mediated by Fenton and Fenton-like reactions involving the reduction of metal ions. Another aspect to consider carefully is the interaction among polyphenols in a mixture. In the case of mixtures of antioxidants, the occurrence of antagonistic or synergistic interactions among the compounds means that the overall antioxidant activity of the mixture does not always correspond to higher values of antioxidant activity of the single components [14,15]. This is particularly important for vegetables and fruits which contain several polyphenols with different degrees of antioxidant and pro-oxidant activities.

These issues have mainly been investigated in in vitro acellular assays [16,17,18], based on cell-free systems. These assays allow a deeper analysis of the antioxidant/pro-oxidant behavior of a polyphenol alone and/or in a mixture and the underlying mechanisms in a wide range of experimental conditions, but they do not provide any information about the capability of a molecule to cross the plasma membrane and about its real behavior inside the cell.

Our work aimed to study the direct intracellular antioxidant/pro-oxidant activity of two known polyphenols, quercetin (3,3′,4′,5,7-pentahydroxyflavone) and resveratrol (3,5,4′-trihydroxytrans-stilbene), alone and in their equimolar binary mixtures in the cell model HeLa cells charged with the cell-permeable ROS-sensitive probe 5-(and-6)-chloromethyl-20,70-dichlorodihydrofluorescein diacetate (CM-H_2_DCFDA) under either basal or pro-oxidant conditions in order to develop a cellular-based short-term assay useful for the cellular assessment of the behavior of antioxidant compounds, singly and in a mixture.

Quercetin and resveratrol are polyphenolic nutraceuticals due to their biological properties. Quercetin is a polyphenolic flavonoid compound in various fruits and vegetables. Several biological activities and health benefits have been ascribed to quercetin, such as antioxidant activity [19], anti-carcinogenic properties [20], antiviral effects [21], antidiabetic activity [22], anti-inflammatory effects [23], anti-aging power [24], angioprotective features [25], neuroprotective activity [26], and anti-obesity activity [27]. Moreover, quercetin is a known apoptotic inducer through the mitochondrial-mediated pathway by activating p53, increasing pro-apoptotic molecules such as Bax, caspase-3, caspase-9,and decreasing anti-apoptotic agents such as survivin and Bcl-2. Resveratrol is a phytoalexin belonging to the stilbene group present naturally in grapes, blackberries, and peanuts as well as medicinal plants such as *Polygonum cuspidatum* [28]. Several biological activities are ascribed to resveratrol such as anti-cancer, anti-inflammatory, and antidiabetic activities, hepato-protection, and inhibition of platelet aggregation [29,30,31,32,33,34]. Moreover, its application as a biologically active ingredient in functional foods has received extensive attention. Quercetin and resveratrol have been recently used in combination in several studies, and a variety of combined effects are reported in the literature [35] such as cell growth inhibition, DNA damage, and cell cycle arrest in cancer cells [36], enhancement of the individual effects of these molecules on triacylglycerol metabolism in white adipose tissue [37], and attenuation of high-fat-diet-induced obesity and associated inflammation in rats via the AMPKα1/SIRT1 signaling pathway [38]. However, despite the extensive research, the direct antioxidant/pro-oxidant behavior of resveratrol and quercetin and their interactions when present in a mixture have mainly been investigated in acellular systems [16,39,40]. To the best of our knowledge, the present work is one of the few studies addressing the direct antioxidant/pro-oxidant activity of resveratrol and quercetin and their interactions in a cellular short-term bioassay focusing on both basal and pro-oxidant conditions.

## 2. Results

### 2.1. Dose-Dependent Effect of Quercetin and Resveratrol Singly and in Mixture on the Basal ROS Production of HeLa Cells

HeLa cells were exposed for 1 h to different concentrations (5, 25, 50 µM) of quercetin, resveratrol, and a mixture of the 2 compounds at equimolar concentrations, and then they were charged with the cell-permeable ROS-sensitive probe CM-H_2_DCFDA to investigate the effects of these compounds and their mixture on the intracellular oxidative status of the cells. Figure 2A–C show the effect size in the different experimental conditions expressed as a percentage variation in the fluorescence emitted by the cells after 1 h exposure calculated with respect to unexposed cells (control). The percentage variation with respect to the control was calculated according to the formula: (treated − control)/control × 100.

Quercetin exerted a dose-dependent antioxidant activity increasing from 5 µM to 50 µM, as indicated by the negative percentage variation in the CM-H_2_DCFDA fluorescence, representative of a decrease in basal ROS production. The antioxidant effect size of quercetin was about 4% at 5 µM, reaching the value of about 20% at 50 µM. Moreover, resveratrol showed a percentage reduction in the ROS basal levels, but the pattern was different. The maximum effect (about 12%) was exerted at the lowest concentration tested (5 µM), while the decrease in the ROS basal production was reduced to 7% for the other 2 concentrations tested (25 and 50 µM). When the cells were exposed to an equimolar mixture of the two compounds, the percentage decrease in the basal ROS production overlapped with the effect observed in the case of quercetin.

### 2.2. Dose-Dependent Effect of Quercetin and Resveratrol, Singly and in Mixture, on Intracellular ROS under An Exogenous Oxidative Challenge

To deepen the analysis of the intracellular antioxidant effects of quercetin and resveratrol single and in a mixture, the cells were exposed to an exogenous oxidative challenge through H_2_O_2_ treatment. As shown in Figure 3, H_2_O_2_ at different concentrations (from 100 µM to 400 µM) hyperbolically increased the intracellular fluorescence of CM-H_2_DCFDA-charged HeLa cells during 1 h observation, with the highest rate of increase observed within the first 10 min. The effect was dose-dependent, with an increase in the fluorescence in the range of 100 µM–300 µM, while a further increase to 400 µM did not exert any significant further fluorescence increase. Therefore, the concentration of 300 µM was chosen for the following experiments under pro-oxidant conditions. The distribution of the probe within the cell was assessed by confocal microscopy to detect the possible presence of any compartmentalization. The confocal visualization of the cells (Figure 2B) showed the intracellular fluorescence homogeneously distributed into the cytoplasm during the H_2_O_2_ exposure with no compartmentation of the dye. This suggests a cytosolic distribution of the probe.

Figure 4 displays the fluorescence time course of HeLa cells preincubated for 1 h with quercetin (A), resveratrol (B), or an equimolar mixture of the 2 polyphenols, loaded with the CM-H_2_DCFDA probe, and then exposed to the oxidative challenge (H_2_O_2_ 300 µM) for 1 h. The initial velocities and the area under the curve of the fluorescence variation of the time courses are reported in Table 1. The H_2_O_2_-induced fluorescence increase was dose-dependently reduced by the preincubation with quercetin (Figure 4A), and either the initial velocities or the area under the curve of the time courses showed a significant dose-dependent reduction (Table 1). These results suggest a protective antioxidant effect of quercetin on the H_2_O_2_-induced oxidative challenge. On the other hand, resveratrol exerted a slight pro-oxidant effect at 5 µM and a more pronounced pro-oxidant effect at 25 and 50 µM, as indicated by the increased values of the initial velocity and the area under the curve. In the case of the mixtures, the observed response was almost overlapping with the quercetin results.

### 2.3. Comparison between Experimental and Expected Antioxidant Activities

The results experimentally obtained with quercetin, resveratrol, and their binary mixtures in HeLa cells in both basal and pro-oxidant conditions (H_2_O_2_ exposure) were compared with the theoretical values calculated by the sum of the effects of the single components at the same concentrations when analyzed separately. This is a common method used by several authors for studying the antioxidant interaction between different compounds [40,41,42,43,44]. Under pro-oxidant conditions, the percentage variation in the fluorescence of the cells after 10 min from the addition of H_2_O_2_ is reported for comparison. As shown in Figure 5, under basal conditions a statistically significant difference between theoretical and experimental values was observed for the three concentrations tested, with the observed values significantly lower than the expected one. This result indicates an antagonistic interaction between the two polyphenols in their mixture.

Under pro-oxidant conditions, at 5 µM no significant difference between theoretical and experimental values was observed, indicating an additive interaction. On the other hand, for the two other concentrations tested, the observed values were significantly higher than the expected one, suggesting a synergistic interaction occurring in the mixture.

## 3. Methods

### 3.1. Materials

All chemicals were reagent-grade. Cell culture materials were acquired from EuroClone (Paignton-Devon, UK). The cell-permeable probe 5-(and-6)-chloromethyl-20,70-dichlorodihydrofluorescein diacetate (CM-H_2_DCFDA) was purchased from Life Technologies-Molecular Probes (Waltham, MA, USA). All the other reagents were purchased from Sigma Aldrich (St. Louis, MO, USA). HeLa cells were purchased from ATCC (Manassas, VA, USA).

### 3.2. Intracellular Antioxidant/Pro-Oxidant Activity Assay

The intracellular antioxidant/pro-oxidant activity of quercetin and resveratrol alone and in their equimolar binary mixtures was assessed in HeLa cells charged with the ROS-sensitive probe, 5-(and-6-)-chloromethyl-20,70-dichlorodihydrofluorescein diacetate, acetyl ester (CM-H_2_DCFDA) (Thermo Fisher Scientific, Waltham, MA, USA).

HeLa cells were grown as a monolayer in Dulbecco’s Modified Eagle’s Medium with 4500 mg glucose/L (DMEM) supplemented with 10% FBS, 40 IU/mL penicillin G, 2 mM L-glutamine, and 100 g/mL streptomycin under a 95% air/5% CO_2_ atmosphere. Cells were plated at a density of 1 × 10^5^ cells per mL into Corning^®^ 96-well solid black flat-bottom polystyrene TC-treated microplates and incubated for 24 h for the adhesion of the cells to the bottom of the plate according to Giordano et al. [45]. For the assessment of the effect of the studied polyphenols on the basal ROS cellular production, HeLa cells were incubated in PBS with different concentrations of quercetin, resveratrol, and equimolar binary mixtures for 1 h. The concentrations of the 2 polyphenols used for single or mixed exposure were 5, 25, and 50 µM, respectively. Then, they were washed and charged with the cell-permeable fluorescent probe CM-H_2_DCFDA 5 µM, which is a chloromethyl derivative of H_2_DCFDA. CM-H_2_DCFDA passively diffuses into cells, where its acetate groups are cleaved by intracellular esterases. Oxidation induces the formation of the fluorescent CM-DCF. Fluorescence was then measured by the Synergy^TM^ (BioTek Instruments, Inc., Winooski, VT, USA) multi-mode microplate reader (Ex/Em: 492–495/517–527 nm).

For the assessment of the effect of the studied polyphenols under pro-oxidant conditions, HeLa cells were first incubated with increasing concentrations of quercetin, resveratrol, and equimolar binary mixtures for 1 h (5, 25, and 50 µM, see above). Then, they were washed, charged with the cell-permeable fluorescent probe CM-H_2_DCFDA 5 µM, and exposed to H_2_O_2_ for 1 h. The time course of the CM-H_2_DCFDA fluorescence variation was recorded by the Synergy MX multi-mode microplate reader (Ex/Em: 492–495/517–527 nm).

### 3.3. Confocal Visualization of HeLa Cells

Cells were plated at a density of 1 × 10^5^ cells per mL into a chambered coverslip (IBIDI, Gräfelfing, Germany), incubated for 24 h for the adhesion of the cells to the bottom of the plate, and then charged with CM-H_2_DCFDA. The cells were viewed under different experimental conditions using a 100× NA plan apochromatic objective mounted on a NIKON TE300 inverted microscope coupled to a NIKON C1 confocal laser scanning unit (Nikon, Tokyo, Japan) [46,47]. The Argon 488 nm laser line was used. Each measurement was performed on at least five fields randomly chosen. Unlabeled cells did not exhibit any detectable fluorescence under the conditions used. Images were acquired and analyzed using EZ-C1 NIKON software.

### 3.4. Statistical Analysis

Data are presented as mean ± S.E.M. For multiple comparisons, analysis was performed by one-way ANOVA followed by Tukey’s multiple comparison post-test, as specified in the captions to figures. GraphPad Prism (version 8) software was used for all the analyses and graphing.

## 4. Discussion

The present work aimed to study the direct intracellular antioxidant/pro-oxidant activity of resveratrol and quercetin and their interactions in a cellular short-term bioassay under both basal and pro-oxidant conditions. The study was carried out by spectrofluorimetric assessment of the intracellular CM-H_2_DCFDA fluorescence in HeLa cells, either natively present in the cells due to the reactive species associated with the normal oxidative metabolism or under pro-oxidant conditions induced by H_2_O_2_ exposure. HeLa cells were chosen as the experimental cell model in this study; they are one of the most studied cellular models, widely utilized in several research fields from cancer research to drug development, gene expression, and cell death pathways, and recently they have been used for the assessment of the antioxidant and pro-oxidant intracellular effects of Trolox, a synthetic analog of vitamin E [45].

H_2_O_2_ was used in this study as an oxidative stress inducer since it is an important physiological oxidative stress agent as a major component of intracellular ROS. It is a membrane-permeable source of the peroxide ion (O_2_^2−^) that is relatively stable in aqueous solutions [48]. Moreover, H_2_O_2_ is known to diffuse through the plasma membrane of mammalian cells through aquaporins [49]. The H_2_O_2_ concentration applied in our experimental system falls within the range of concentrations found extracellularly during pathological conditions such as inflammation and tumor growth [50]. The cellular assay was based on the ROS-sensitive intracellular probe CM-H_2_DCFDA. In our experimental system, its fluorescence increased hyperbolically in a dose-dependent manner under exposure to H_2_O_2_. As outlined by Kalyanaraman et al. [51], the oxidation of this sensor is not specific to a particular type of ROS; therefore, it does not allow a direct measure of intracellular H_2_O_2_ since the probe can be oxidized not only by H_2_O_2_ but by several oxidizing species including hydroxyl radicals (·OH), compounds I and II generated from the interaction between H_2_O_2_ and peroxidase or heme, NO_2_ arising from the myeloperoxidase/H_2_O_2_/NO_2_ system, and reactive species arising from peroxynitrite (ONOO-/ONOOH) decomposition. Therefore, the hyperbolic increase in the fluorescence of the probe following exposure of the cells to H_2_O_2_ fluorescence does not directly reflect a time-dependent increase in the H_2_O_2_ intracellular concentration, but it somewhat represents the appearance of a time-dependent increase in the H_2_O_2_-induced intracellular oxidative conditions.

Under basal conditions, both quercetin and resveratrol exerted an antioxidant behavior against the ROS basal levels arising from cellular oxidative metabolism but with some differences. The effect was more pronounced in the case of quercetin which showed a dose-dependent antioxidant behavior; its effect was statistically significant at already 5 µM and further increased at 25 µM and 50 µM, reaching the maximum value of about 20% CM-H_2_DCFDA fluorescence percentage variation. In the case of resveratrol, the antioxidant effect was weaker, with a maximum value (about 7%) at the lowest concentration tested of 5 µM. This result suggests a more effective ability of quercetin to exert a direct antioxidant activity against ROS arising from the basal cellular metabolism, presumably due to the known antiradical activity of the catechol group (3′,4′-dihydroxy) in the B ring and the double bond between carbon 2 and carbon 3 of the C ring conjugated with the keto group at position 4 [9]. On the other hand, the direct antioxidant behavior of resveratrol against the intracellular ROS basal levels was weaker. The direct antioxidant activity of resveratrol has been widely assessed in different cell-free assays including total antioxidant activity, reducing power, DPPH^•^, ABTS^•+^, DMPD^•+^ and O_2_^•−^ radical scavenging, hydrogen peroxide scavenging, and metal chelating activities [52]. Resveratrol has been shown to directly scavenge a variety of oxidants, including hydroxyl radical (^●^OH), O_2_^●−^, H_2_O_2_, and peroxynitrite in acellular assays. However, its direct ROS scavenging activity appears to be poor [53,54], while it is thought that the known effects of resveratrol against oxidative injury in vivo are more likely to be attributable to its effects as a gene regulator of antioxidant enzymes rather than to its direct scavenging activity. In our system, the short duration of the assay, used to investigate the direct antioxidant activity intracellularly, prevented the expression of the indirect antioxidant activity of resveratrol as a modulator of antioxidant enzyme expression and could explain the weak antioxidant effect observed.

In our system, under pro-oxidant conditions, quercetin exerted a significant antioxidant activity, as indicated by the significant (*p* < 0.05) decrease in the initial velocity (calculated as the first derivative *f’* at time = 0) and area under the curve of the fluorescence variation time course of HeLa exposed to 300 µM H_2_O_2_ cells. It acted extremely fast in the first few minutes, as indicated by the significantly (*p* < 0.05) decreased initial velocity of the time course. Indeed, the antioxidant activity was significantly detectable at 5 µM and reached its maximum value at 25 µM, while at 50 µM no further increase in the antioxidant activity was observed. These results obtained in a cellular bioassay are in agreement with the well-known capacity of quercetin to lower ROS formation by scavenging activity, as assessed in cell-free assays [9,55], and outline the direct intracellular ROS scavenging activity of this polyphenol.

On the other hand, in the same cell-based assay, resveratrol exerted a pro-oxidant behavior under exposure of the HeLa cells to H_2_O_2_, as indicated by the significant increase in the initial velocity and area under the curve. The effect was statistically significant at higher concentrations starting from 25 µM, while it was not detectable at the lowest concentration tested of 5 µM. It is known that resveratrol may behave as an antioxidant or pro-oxidant depending on concentration, time of exposure, cell type, and the presence of metal cations [56,57]. This, in turn, influences the biological effect exerted, as observed in previous studies where cardioprotective properties of resveratrol were found at lower concentrations (5 μM–10 μM) associated with an antioxidant behavior, while at higher concentrations it acted as a pro-oxidant [58]. Moreover, at low concentrations (5 μM) resveratrol increased cell proliferation, while at higher concentrations (15 μM or more) it induced apoptosis in various cancer cells [59]. It is known that resveratrol can be (auto-)oxidized to generate semiquinones and the relatively stable 4′-phenoxyl radical, in turn leading to ROS production [60,61,62]. Such polyphenol oxidative reactions are known to be influenced by pH and the presence of hydroxyl anions [63,64]. The exposure of the cells to H_2_O_2_ increased the intracellular concentrations of hydroxyl anions [65]; therefore, this could explain the pro-oxidant effect observed in our system under exposure to H_2_O_2_.

In our cellular-based assay, the possible interaction between the two polyphenols was investigated in both basal and pro-oxidant conditions. It is known that the assessment of the antioxidant activity of the single components of a mixture does not allow determining the total antioxidant capacity of the mixture due to the interaction of the single compounds [40]. We compared the experimental results obtained with the binary mixtures of quercetin and resveratrol at the three concentrations tested with the theoretical values calculated by the algebraic sum of the effects of the single components at the same concentrations, according to previous studies [40,41,42,43,44]. Under basal conditions, an antagonistic interaction between quercetin and resveratrol in equimolar binary mixtures was found at the three concentrations tested. Indeed, the sum of the effects, both attributable to an antioxidant behavior, was less than the mathematical sum that would be predicted from the individual components according to Wang et al. [29]. On the other hand, under pro-oxidant conditions where opposite effects were observed (antioxidants for quercetin and pro-oxidants for resveratrol), different results were observed at different concentrations. An additive interaction was found at the lowest concentration tested (5 µM) while a synergic effect was observed at the higher concentrations tested, where the pro-oxidant effect of resveratrol was completely nullified by the presence of quercetin. In this case, quercetin activity seems to be enhanced by the pro-oxidant effect of resveratrol compared with the baseline conditions. In the literature, little information is available on the interaction of quercetin and resveratrol under direct antioxidant activity determination, and no information is available in the intracellular environment. Skroza et al. [39] previously analyzed the behavior of quercetin and resveratrol equimolar binary mixtures using the cell-free antioxidant assay Briggs–Rauscher reaction, finding slight antagonism. The results obtained with HeLa cells appear different from the data obtained in cell-free assays, suggesting that the interaction of polyphenols in the intracellular environment cannot be predicted by their behavior in a cell-free reaction; moreover, the oxidative status of the cells and concentrations seem to strongly influence the type of interactions.

## 5. Conclusions

In conclusion, this study clarified the direct intracellular antioxidant/pro-oxidant activity of two known polyphenols, quercetin and resveratrol, alone and in their equimolar mixtures on the cell model HeLa cells charged with the cell-permeable ROS-sensitive CM-H_2_DCFDA under either basal or pro-oxidant conditions. The results obtained showed under basal conditions a significant antioxidant effect of quercetin and a weaker antioxidant effect of resveratrol when used singly, while antagonism of their effect was detected in their equimolar mixtures at all the concentrations used. Under pro-oxidant conditions, exerted by the exposure of the cells to H_2_O_2_, quercetin exhibited dose-dependent intracellular antioxidant activity, whereas resveratrol manifested a pro-oxidant intracellular activity, while their equimolar mixtures highlighted an intracellular interaction between the 2 polyphenols, with additive effects at 5 µM and synergic at 25 µM and 50 µM, which were not predictable on the basis of the analysis conducted on the single compounds. Thus, the results highlight that all the intracellular antioxidant properties of polyphenols in mixtures depend not only on the nature of the compounds themselves but also on the type of interaction in the cellular system and the oxidative status of the cells. Moreover, in the cellular system, the behavior of polyphenols singly and in mixtures does not always correspond with an ability to counteract the production of ROS in cell-free systems. The results can be used as a model for further studies with other polyphenols and cell types; they can represent a base for the development of a cellular-based short-term assay useful for the assessment of the intracellular behavior of antioxidant compounds, singly and in a mixture.

## Figures and Tables

**Figure 1 molecules-28-04682-f001:**
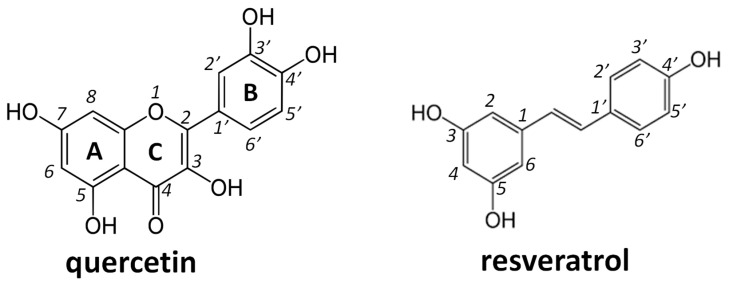
Structural formula of quercetin and resveratrol.

**Figure 2 molecules-28-04682-f002:**
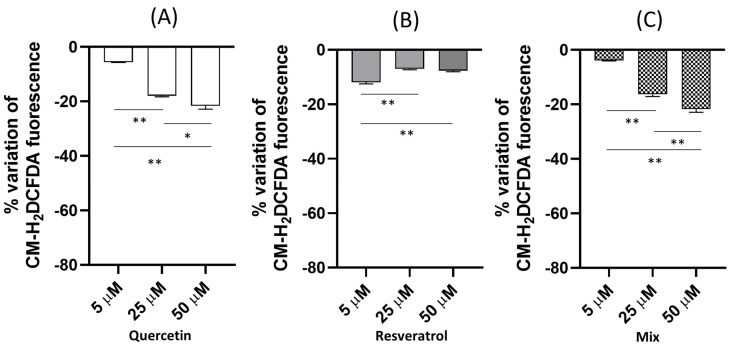
(**A**–**C**) Effect of increasing concentrations of quercetin, resveratrol, and equimolar mixtures of the two compounds on the basal fluorescence of HeLa cells loaded with the CM-H_2_DCFDA probe. The effect was measured spectrofluorimetrically after 1 h incubation and is reported as a % variation in the CM-H_2_DCFDA calculated with respect to the control (unexposed cells). The statistical significance of the data was analyzed by one-way ANOVA and Tukey’s multiple comparison test. Data are expressed as mean ± SEM. ** *p* < 0.01; * *p* < 0.05.

**Figure 3 molecules-28-04682-f003:**
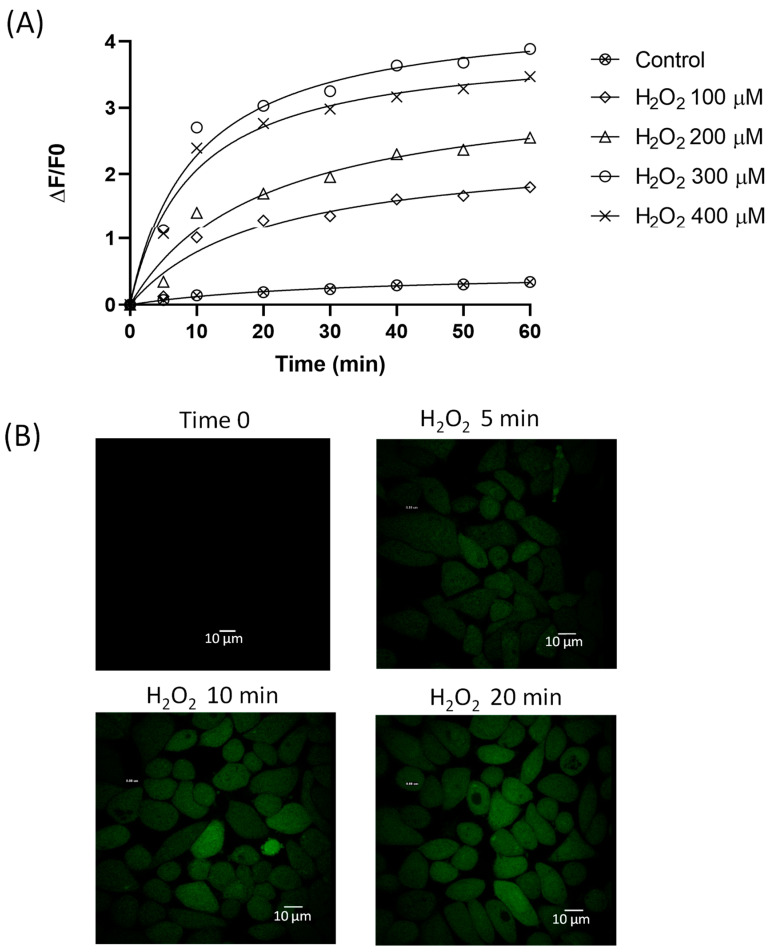
(**A**,**B**) Effect of increasing concentrations of H_2_O_2_ on the fluorescence of HeLa cells loaded with the CM-H_2_DCFDA probe. The effect was measured spectrofluorimetrically over 1 h. Control refers to cells loaded with the probe but not exposed to the peroxide. Data are expressed as mean ± SEM. (**B**) Representative confocal images of HeLa cells charged with the probe before and after the exposure to 300 µM H_2_O_2_.

**Figure 4 molecules-28-04682-f004:**
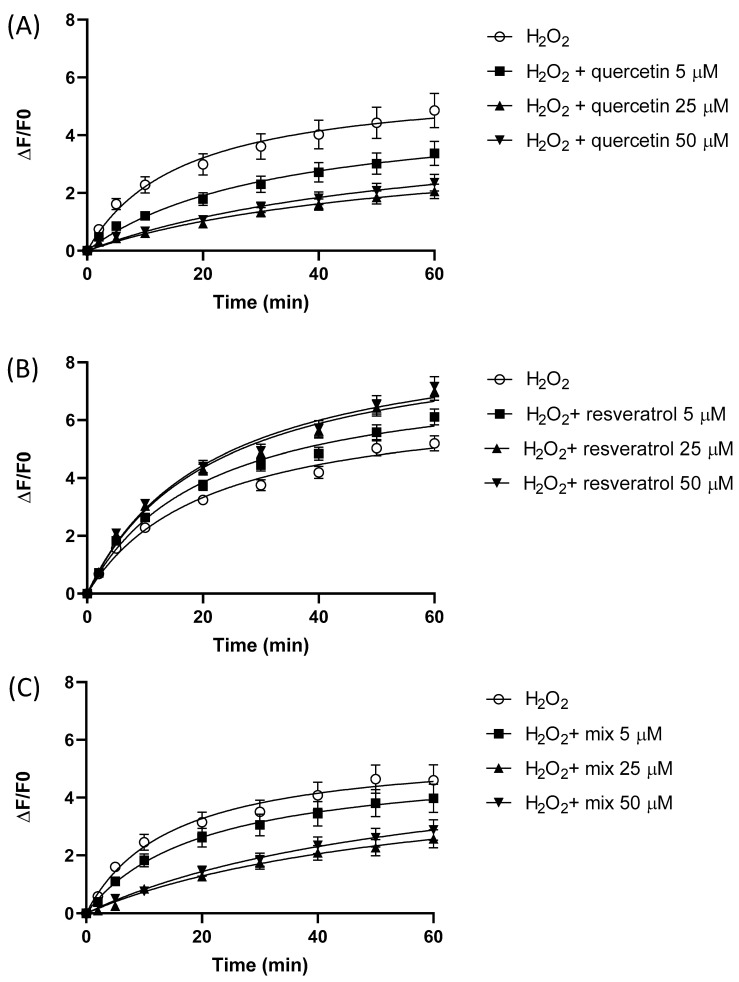
(**A**–**C**) Effect of 1 h preincubation with increasing concentrations of quercetin (**A**), resveratrol (**B**), and an equimolar mixture of the 2 polyphenols on the fluorescence variation time course of HeLa cells loaded with the CM-H_2_DCFDA probe and exposed to an oxidative challenge (H_2_O_2_ 300 µM). The effect was measured spectrofluorimetrically over 1 h. Data are expressed as mean ± SEM.

**Figure 5 molecules-28-04682-f005:**
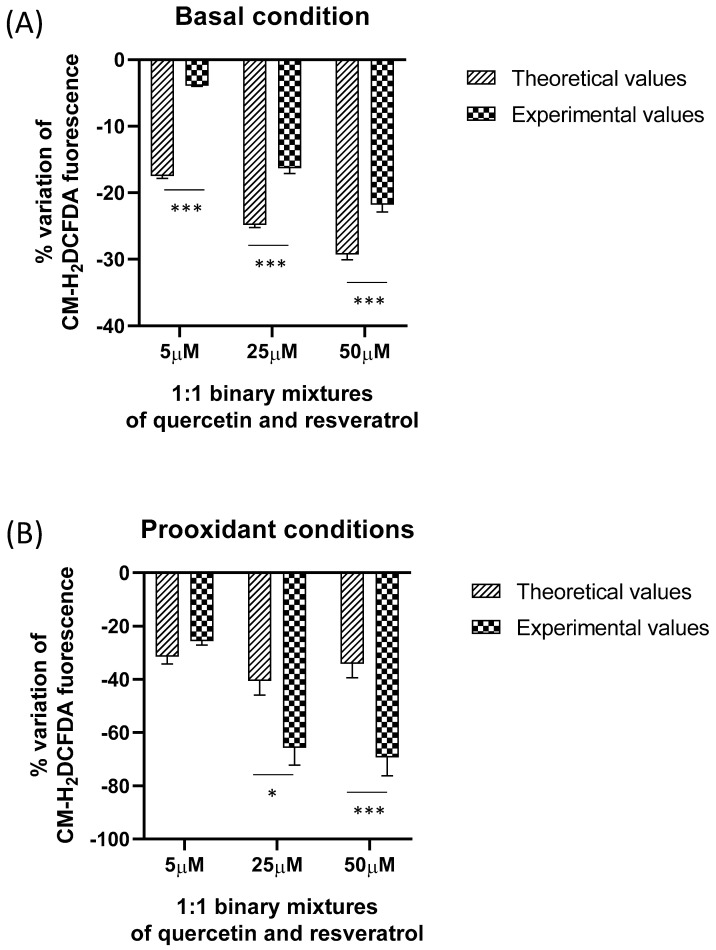
(**A**,**B**) Comparison between the activity (expressed as percentage variation of the control) of equimolar mixtures of resveratrol and quercetin measured experimentally in basal (**A**) and pro-oxidant conditions (**B**) with the corresponding theoretical values calculated by adding up the effect of the individual components at the same concentration analyzed separately. Data are expressed as mean ± SEM. * *p* < 0.05; *** *p* < 0.001.

**Table 1 molecules-28-04682-t001:** Initial velocity (calculated as the first derivative *f*’ at time = 0) and area under the curve of the fluorescence variation time course of HeLa cells loaded with the CM-H_2_DCFDA probe and exposed to an oxidative challenge (300 µM H_2_O_2_). The cells were preincubated with different concentrations of quercetin, resveratrol, or an equimolar mixture of the two polyphenols. Data were analyzed by one-way ANOVA and Tukey’s multiple comparison test. Different letters are used to show statistical significance (*p* < 0.05).

	Initial Velocity (ΔF/F_0_ x min^−1^)	Area under Curve (AUC)
300 µM H_2_O_2_	0.356 ± 0.017 ^a^	200.3 ± 7.4 ^a^
300 µM H_2_O_2_ + 5 µM quercetin	0.154 ± 0.008 ^b^	128.9 ± 4.9 ^b^
300 µM H_2_O_2_ + 25 µM quercetin	0.069 ± 0.004 ^c^	74.40 ± 2.9 ^c^
300 µM H_2_O_2_ + 50 µM quercetin	0.077 ± 0.004 ^c^	84.39 ± 3.2 ^c^
300 µM H_2_O_2_	0.326 ± 0.020 ^a^	213.3 ± 7.9 ^a^
300 µM H_2_O_2_ + 5 µM resveratrol	0.382 ± 0.019 ^a,b^	245.7 ± 8.4 ^a,b^
300 µM H_2_O_2_ + 25 µM resveratrol	0.429 ± 0.022 ^b^	279.9 ± 9.2 ^b^
300 µM H_2_O_2_ + 50 µM resveratrol	0.428 ± 0.018 ^b^	285.0 ± 10.4 ^b^
300 µM H_2_O_2_	0.386 ± 0.022 ^a^	203.2 ± 6.8 ^a^
300 µM H_2_O_2_ + 5 µM mix	0.263 ± 0.015 ^b^	168.3 ± 6.3 ^b^
300 µM H_2_O_2_ + 25 µM mix	0.086 ± 0.006 ^c^	94.2 ± 3.7 ^c^
300 µM H_2_O_2_ + 50 µM mix	0.096 ± 0.005 ^c^	105.8 ± 4.1 ^c^

## Data Availability

The authors confirm that the data supporting the findings of this study are available within the article.

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
