# Peer review of "Intracellular Redox Behavior of Quercetin and Resveratrol Singly and in Mixtures"

_molecules, 2023, doi:10.3390/molecules28124682_

Round 1

Reviewer 1 Report

(1) The author submitted a manuscript with the research direction of food chemistry, which is described in the preface as molecular biology and cell biology, why? Is it possible to add the research progress related to food chemistry.

2) Line 96, there is an error in the format of references.

3) Suggest to streamline the number of keywords.

4) It is suggested to add the information of HeLa cells, not limited to the process of your activation.

Language and and article formatting issues require partial revision and careful examination. This does not mean that you have a very serious problem, just that there is a relevant problem that needs you to make the research paper perfect.

Author Response

The authors would like to thank the reviewer for his/her valuable comments. His/her input has been very helpful in improving the manuscript. Please, see our responses below. The changes made in the text are reported in Track Changes in the manuscript and clearly described in the answers to the referee’s comments.

1) The author submitted a manuscript with the research direction of food chemistry, which is described in the preface as molecular biology and cell biology, why? Is it possible to add the research progress related to food chemistry.

            Response: The choice of the section food chemistry arises from the fact that in recent years polyphenols have catalyzed great interest in the research field of functional foods and nutraceuticals within the wide research area of food chemistry. Therefore, according to the referee’s suggestion, the following sentence was added in the introduction (pag. 4): “In particular, polyphenols have attracted great interest in the last decades as potent antioxidant compounds; they have catalyzed great interest in the research field of functional foods and nutraceuticals within the wide research area of food chemistry”.

2) Line 96, there is an error in the format of references.

Response: the error has been corrected

3) Suggest to streamline the number of keywords.

Response: According to the referee’s suggestion the number of keywords has been streamlined

4) It is suggested to add the information of HeLa cells, not limited to the process of your activation.

Response: According to the Referee’s suggestion more information on HeLa cells was added in the discussion as follows “HeLa cells were chosen as the experimental cell model; they are one of the most studied cellular models, widely utilized in several research fields from cancer research, to drug development, gene expression, cell death pathways, and recently for the assessment of antioxidant and prooxidant intracellular effects of Trolox, a synthetic analog of vitamin E [42]”.

 Language and article formatting issues require partial revision and careful examination. This does not mean that you have a very serious problem, just that there is a relevant problem that needs you to make the research paper perfect.

Response: According to the referee’s suggestion language and article formatting have been revised

Reviewer 2 Report

The Authors investigate the direct intracellular antioxidant or prooxidant activity of quercetin and resveratrol, hence their interaction, in a cellular short term bioassay within a basal and a prooxidant condition.

I find the study properly designed and the results efficiently discussed; yet, it could be helpful to include the chemical structure of the polyphenols mentioned in the introduction, as already effectively described. 

Furthermore, legends of the figures could be more clear and organized with the capital letter inserted next to the object, the scale bar of the confocal images and the statistical significance of letters in Table 1; besides, the discussion of Table 1 could be more precise.

Please double check the unit M in Fig.2 (A) and the text in the discussion written with a different character. 

Author Response

The authors would like to thank the reviewer for his/her valuable comments. His/her input has been very helpful in improving the manuscript. Please, see our responses below. The changes made in the text are reported in Track Changes in the manuscript and clearly described below.

Referee: The Authors investigate the direct intracellular antioxidant or prooxidant activity of quercetin and resveratrol, hence their interaction, in a cellular short term bioassay within a basal and a prooxidant condition.

I find the study properly designed and the results efficiently discussed; yet, it could be helpful to include the chemical structure of the polyphenols mentioned in the introduction, as already effectively described. 

Response: According to the referee’s suggestion the chemical structure of the polyphenols mentioned in the introduction has been included in the new Fig.1

Referee: Furthermore, legends of the figures could be more clear and organized with the capital letter inserted next to the object, the scale bar of the confocal images and the statistical significance of letters in Table 1; besides, the discussion of Table 1 could be more precise.

Response: According to the referee’s suggestion legends of the figures were better organized with the capital letter inserted next to the object, the scale bar of the confocal images was added, and the statistical significance of letters was added in the caption to Table 1. Moreover, the discussion of Table 1 was improved. In the result section (pag.11) the sentenceQuercetin exerted a protective antioxidant effect on the H2O2 oxidative challenge since the H2O2-induced fluorescence increase was dose-dependently reduced by the preincubation with quercetin (Fig.3A), as indicated by the reduction of the initial velocities of the time-courses and the reduction of the area under the curve (Tab. 1). On the other hand, resveratrol exerted a slight prooxidant effect at 5 µM and a more pronounced prooxidant effect at 25 and 50 µMwas changed toThe H2O2-induced fluorescence increase was dose-dependently reduced by the preincubation with quercetin (Fig.4A) and either the initial velocities or the area under curve of the time-courses showed a significant dose-dependent reduction (Tab.1). These results suggest a protective antioxidant effect of quercetin on the H2O2 induced oxidative challenge. On the other hand, resveratrol exerted a slight prooxidant effect at 5 µM and a more pronounced prooxidant effect at 25 and 50 µM as indicated by the increased values of the initial velocity and the area under curve.”Moreover, in the discussion section (pag.15) the sentenceIn our system, under prooxidant conditions, quercetin exerted a significant antioxidant activity as indicated by both the initial velocity (calculated as the first derivative f’ at time = 0) and area under the curve of the fluorescence variation time course of HeLa exposed to 300 µM H2O2. It acted extremely fast in the first few minutes as indicated by the initial velocity of the time course. Indeed, the antioxidant activity was significantly detectable at 5µM and reached its maximum value at 25 µM, while at 50 µM no further increase of the antioxidant activity was observed.” was changed toIn our system, under prooxidant conditions, quercetin exerted a significant antioxidant activity as indicated by the significant (p<0.05) decrease of the initial velocity (calculated as the first derivative f’ at time = 0) and area under the curve of the fluorescence variation time course of HeLa exposed to 300 µM H2O2 cells. It acted extremely fast in the first few minutes as indicated by the significantly (p<0.05) decreased initial velocity of the time course. Indeed, the antioxidant activity was significantly detectable at 5µM and reached its maximum value at 25 µM, while at 50 µM no further increase of the antioxidant activity was observed”.

Referee: Please double check the unit M in Fig.2 (A) and the text in the discussion written with a different character. 

Response: According to the referee’s suggestion the unit “M” was changed to “µM” e the character was revised.